# Interactive Repair for Robust Generative Manipulation with Sparse Human Nudges

Anonymous

*Abstract*— **Learned manipulation policies remain brittle under real-world distribution shift, often failing on narrow but important edge cases despite strong performance on nominal examples. Many such failures are near-misses: the robot reaches an almost-correct state, and a small corrective motion could recover the task. We propose an interactive repair method that converts sparse human nudges into localized updates of a pretrained generative manipulation policy without retraining the backbone. The method keeps the base policy frozen and learns a lightweight repair module that adjusts behavior only in corrected regions, helping preserve existing competence outside the failure regime. We evaluate the approach on a real robot across tabletop manipulation tasks involving grasping, pouring, cup uprighting, and insertion under hard in-distribution and out-of-distribution conditions. With a small correction budget, the repaired policy recovers many previously failed cases while maintaining or improving broader task performance. Compared with full retraining on the same correction data, interactive repair offers a more localized and compute-efficient path to improving robustness in deployment.**

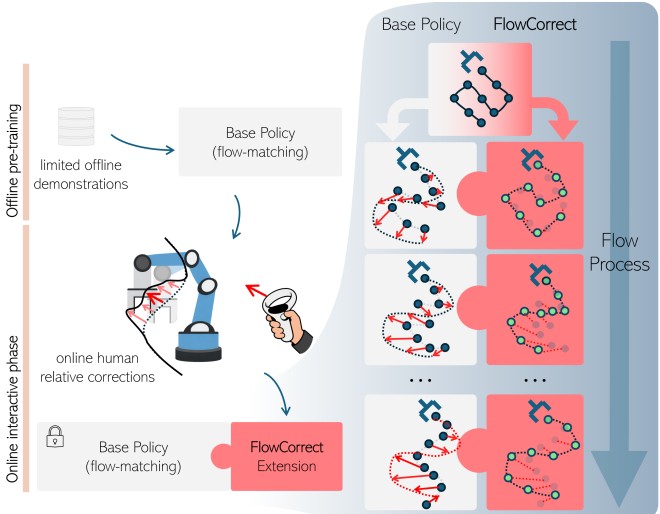

Fig. 1. Overview of our interactive repair framework (*FlowCorrect*). A pretrained generative manipulation policy runs at deployment time while a human provides occasional relative corrective nudges. These sparse corrections are used to train a lightweight repair module that locally adjusts the policy without retraining the backbone.

## I. INTRODUCTION

Robustness remains a central challenge in robot manipulation. Learned policies can perform well on nominal demonstrations and benchmark settings, yet fail under small real-world variations in object pose, geometry, contact conditions, or scene layout. In many cases these failures are not complete breakdowns but near-misses: the robot reaches an almost-correct state, and a small adjustment would recover the task. This suggests that improving robustness is not only a matter of larger training sets or broader pretraining, but also of enabling targeted repair when failures are encountered during deployment.

A common way to address such failures is to fine-tune or retrain the policy on additional data [1], but this can be expensive and may unintentionally alter behavior in previously successful regions. For deployment-time robustness, we instead seek localized updates that repair brittle failure modes without overwriting existing competence.

We explore this through sparse relative human nudges: instead of taking over control and providing a full corrective trajectory, a human supplies a small motion adjustment to the robot's current behavior. This interactive repair setting is closely related to interactive imitation learning, where supervision is provided during deployment rather than only offline training [2]. We use these corrections to learn a lightweight repair module on top of a frozen generative policy, enabling interactive policy repair without full retraining (see Fig. 1).

To summarize, our contributions are as follows:

- We present an interactive repair method for improving robustness of pretrained generative manipulation policies under real-world failures.
- We introduce a lightweight local repair module that converts sparse human nudges into policy edits while preserving the frozen backbone.
- We validate the approach on real-robot tabletop manipulation tasks and compare it to full retraining under low correction budgets.

## II. RELATED WORK

Prior interactive robot learning approaches often rely on full teleoperation, absolute action corrections, or retraining on corrected trajectories [3]–[6]. Residual and modular adaptation methods reduce forgetting by learning updates on top of a base policy, but are typically not formulated as localized edits to a generative action field [7], [8]. Our work combines these directions: we target pretrained generative manipulation policies and repair them at deployment time from sparse relative human corrections using a lightweight local adapter.

## III. BACKGROUND AND PROBLEM SETTING

We consider a pretrained manipulation policy that maps recent observations to a short horizon of end-effector actions. The policy is trained from offline teleoperated demonstrations and deployed on a real robot. At deployment time, a human observer can intervene on difficult executions by supplying sparse relative corrective nudges, which we use to repair the policy locally.

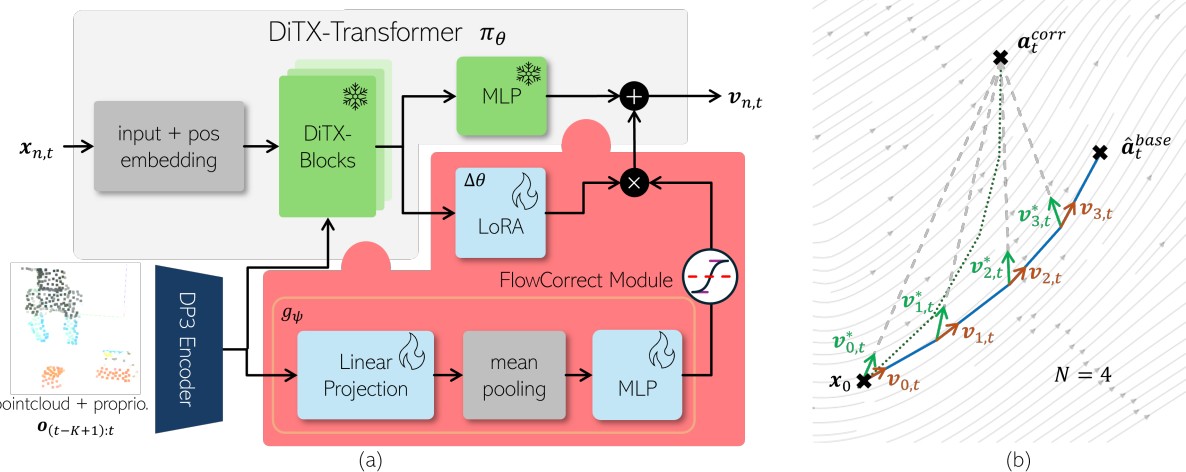

(a)                                                                                      (b)

Fig. 2.   (a) Overview of *FlowCorrect* module that is attached to the DiTX-Transformer from ManiFlow [9]: we extend an existing flow matching policy $\pi_\theta$ based on DiTX-Transformer with our *FlowCorrect* module. Our lightweight *FlowCorrect* module consists of LoRA adapters (parametrized by $\Delta\theta$) injected into the transformer, and a gating module $g_\psi$ that outputs a signal to steer the vector flow field towards the corrected action. (b) Intuition: across N=4 integration steps, *FlowCorrect* iteratively adjusts the predicted velocities from $v_{n,t}$ to $v_{n,t}^*$, steering the rollout from a base action $\hat{a}_t^{\text{base}}$ toward a corrected action $a_t^{\text{corr}}$.

Our base policy is a flow-based generative policy that predicts an action chunk by iteratively integrating a learned vector field $f_\theta$ conditioned on recent observations. Flow-Correct leaves this pretrained backbone frozen and learns a lightweight additive correction to this vector field from sparse human corrections.

## IV. METHODOLOGY

Our method repairs a pretrained generative policy by adding a lightweight adapter while keeping the original backbone frozen. During deployment, a human provides occasional corrective nudges on failing executions. These sparse corrections are then used to update only the repair module, yielding a locally adapted policy.

The goal of interactive repair is to make the adapted policy follow human-corrected actions on failure cases while staying close to the pretrained policy elsewhere. Because corrections are sparse, the update must be parameter-efficient and localized, so that previously successful behaviors are preserved.

### A. Sparse repair signals from human nudges

Rather than asking a human to teleoperate the robot and provide full replacement actions, we use sparse relative corrections. At corrected timesteps, the human supplies an offset $\mathbf{b}_t$ to the robot's current action prediction, yielding

$$a_t^{\text{corr}} = \hat{a}_t^{\text{base}} \oplus \boldsymbol{b}_t$$

This makes corrections a compact signal for rapid repair of near-miss failures.

In practice, the human applies these nudges through a lightweight VR-based interface that produces smooth relative pose adjustments and gripper corrections during execution. We log the resulting corrected trajectories together with the corresponding observations and base-policy predictions, and use them as repair data.

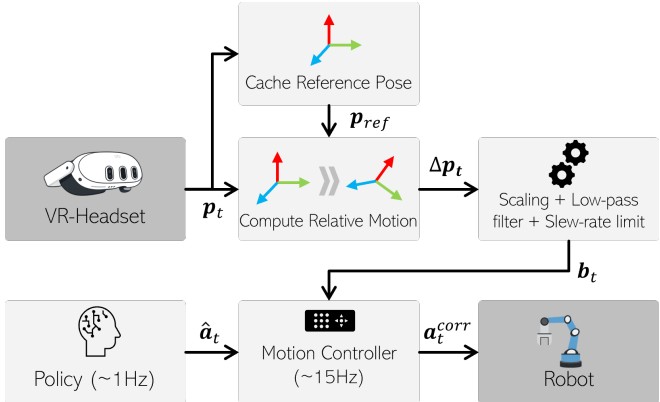

Fig. 3.   Pipeline of the interactive correction interface.

### B. Local repair of the generative policy

We repair the pretrained policy by learning a lightweight additive update to its vector field. Concretely, we attach a small LoRA-based adapter to the frozen backbone so that the repaired vector field becomes

$$f_{\theta+\Delta\theta}(\boldsymbol{x}_n, k_n, \boldsymbol{c}) = f_\theta(\boldsymbol{x}_n, k_n, \boldsymbol{c}) + \alpha_t \boldsymbol{v}_{\Delta\theta}(\boldsymbol{x}_n, k_n, \boldsymbol{c}).$$

where $f_\theta$ is the pretrained field, $\boldsymbol{v}_{\Delta\theta}$ is the learned repair term, and $\alpha_t$ is a gate that determines where the repair should be applied. At inference, the gate suppresses the repair outside corrected regions, helping preserve the base policy on nominal states.

At each integration step, we define a target velocity that would steer the current latent action state toward the corrected action by the end of integration:

$$\boldsymbol{v}_{n,t}^* = \frac{\boldsymbol{a}_t^{\text{corr}} \ominus \boldsymbol{x}_{n,t}}{(N-n)\Delta k},$$

The *FlowCorrect* loss for a single flow trajectory is

$$\mathcal{L}_{\text{FE}}(\Delta\theta) = \frac{1}{N} \sum_{n=0}^{N-1} w_n \left\| f_{\theta+\Delta\theta}(\boldsymbol{x}_n, k_n, \boldsymbol{c})_t - \boldsymbol{v}_{n,t}^* \right\|_2^2.$$

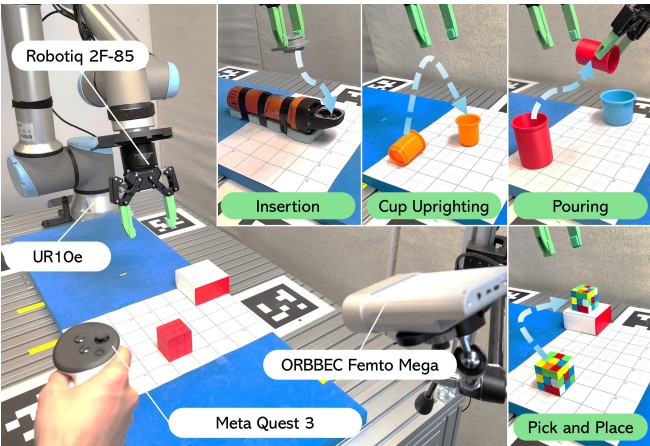

Fig. 4. Hardware setup and representative real-world tasks used in the experiments.

This loss shapes the local action flow so that the repaired rollout converges toward the corrected execution rather than simply fine-tuning on corrected terminal actions.

### C. Preserving robustness outside repaired regions

Local repair is only useful if it does not reduce robustness elsewhere. We therefore use two preservation mechanisms: a gate that activates the repair only in relevant regions, and a small set of successful rollout anchors that regularize the repaired policy toward the original behavior outside the corrected failure modes.

## V. EXPERIMENTS

We evaluate whether sparse human nudges can repair hard deployment-time failures while preserving prior behavior better than full retraining.

We report success rates over a structured set of in-distribution (ID) and out-of-distribution (OOD) initial conditions, and perform ablations to isolate the contributions of gating and rollout data.

### A. Setup

We evaluate on a UR10 robot with a parallel-jaw gripper across four tabletop tasks: pick-and-place, pouring, cup uprighting, and insertion (Fig. 4). A pretrained generative policy is learned from eight teleoperated demonstrations per task. For each selected hard case, we collect ten corrected rollouts from sparse human nudges and five successful uncorrected rollout anchors. We compare the **Base** policy, our repaired **FC** policy, and full-policy retraining (**RT**) using the same correction data.

### B. Results

*FlowCorrect* reliably solves hard ID and OOD cases in Cup Uprighting and all selected ID-hard cases in Pouring, but transfers only marginally to an OOD height change. In Pick-and-Place, it greatly improves most hard cases, yet one nearby ID-hard setting improves only partially, suggesting that local correction can overfit to one solution when nearby states require conflicting edits (Table I).

TABLE I

STRESS-TEST SUCCESS ON SELECTED HARD POSITIONS. "ID-HARD" ARE THE THREE LOW-PERFORMING ID CONDITIONS; "OOD-HARD" IS THE SELECTED OOD CONDITION.

| Task | Policy | ID-hard1 | ID-hard2 | ID-hard3 | OOD-hard | Σ |
|------|--------|------|------|------|------|------|
| Pick-and-Place | **Base** | 0/10 | 0/10 | 0/10 | 0/10 | 0/40 |
| | **FC** (Ours) | 3/10 | **10**/10 | 9/10 | **10**/10 | 32/40 |
| | **RT** | **10**/10 | **10**/10 | **10**/10 | **10**/10 | 40/40 |
| Pouring | **Base** | 4/10 | 0/10 | 0/10 | 0/10 | 4/40 |
| | **FC** (Ours) | **10**/10 | **10**/10 | **10**/10 | 2/10 | 32/40 |
| | **RT** | **10**/10 | **10**/10 | **10**/10 | **10**/10 | 40/40 |
| Cup Uprighting | **Base** | 0/10 | 0/10 | 0/10 | 0/10 | 0/40 |
| | **FC** (Ours) | **10**/10 | **9**/10 | **10**/10 | **9**/10 | 38/40 |
| | **RT** | **10**/10 | 8/10 | 8/10 | **9**/10 | 35/40 |
| Insertion | **Base** | 0/10 | 0/10 | 0/10 | 0/10 | 0/40 |
| | **FC** (Ours) | **10**/10 | 2/10 | **10**/10 | **10**/10 | 32/40 |
| | **RT** | 8/10 | **8**/10 | 7/10 | 9/10 | 32/40 |

Figure 5 shows the different ID and OOD conditions as well as some common failure cases of the base policy before adaptation with *FlowCorrect*. Typical failure cases include an unstable grasp, collision with the object, or misalignment with the object. Each ID and OOD condition is evaluated 10 times to define a success rate.

Across 30 ID initial conditions, *FlowCorrect* improves the base policy on all four tasks, showing that sparse corrections can stabilize execution beyond the corrected timesteps. Gains are largest in Pouring and Cup Uprighting, while improvements are smaller in Pick-and-Place and Insertion, where success depends on more precise gripper placement (see Fig. 6).

Retraining performs strongly on hard cases, but *FlowCorrect* remains competitive in overall ID success while updating only a small LoRA module and lightweight gate. In Insertion, retraining notably reduces overall ID success, suggesting that full-model updates can disrupt previously successful behaviors in high-precision tasks. It also requires substantially more training time and GPU memory, making *FlowCorrect* the more efficient adaptation method.

In additional ablations, we found that gating and successful rollout anchors improved preservation of in-distribution behavior.

TABLE II

RESOURCE USAGE COMPARISON BETWEEN OUR *FlowCorrect* TRAINING (**FC**) AND RETRAINING (**RT**).

| Mode | Avg. GPU Memory Usage (GB) | Avg. Runtime (min.) |
|------|------|------|
| **Base** | 18.84±0.24 | 80.86±10.01 |
| **FC** (Ours) | **4.35±0.15** | **30.24±5.45** |
| **RT** | 19.23±0.25 | 52.93±10.96 |

### C. Discussion and Outlook.

Our experiments also reveal two limitations. First, when nearby situations require conflicting local repairs, a single lightweight edit can interfere across cases. Second, shifts driven by object geometry are harder to repair than pose-centric near-misses, suggesting that richer observation-

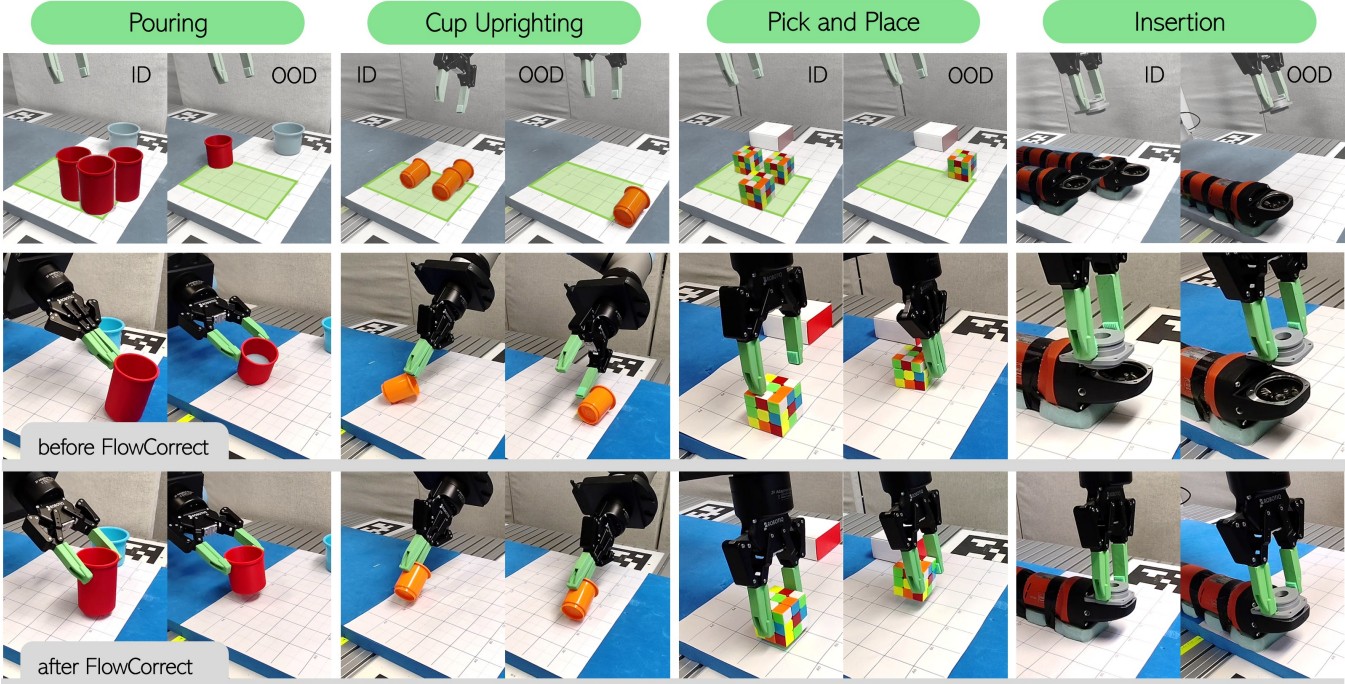

Fig. 5. Top row: Selected ID-hard and OOD-hard initial conditions for the four tasks (left to right): Pouring, Cup Uprighting, Pick-and-Place, and Insertion. The green regions indicate the workspace areas covered by the demonstrations. Middle row: Representative failure cases of the base policy under these conditions. Bottom row: Qualitative examples of successful executions after *FlowCorrect* fine-tuning on conditions that previously failed.

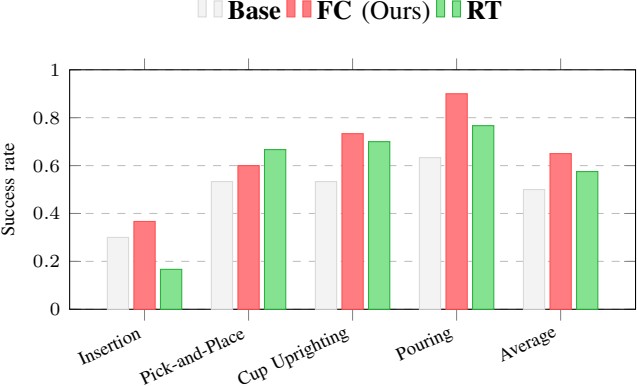

Fig. 6. Overall success rate across 30 ID positions in the workspace. Gray denotes the base policy; red denotes our *FlowCorrect* (FC) policy after correcting only three ID conditions; and green denotes the base policy retrained (RT) with the same three ID corrections. Notably, *FlowCorrect* improves the original performance without regression.

conditioned routing may be needed. Addressing these limitations is an important direction for future work.

## VI. CONCLUSIONS

We presented interactive repair as a practical mechanism for improving robustness of generative manipulation policies under real-world failures. By learning a lightweight local repair module on top of a frozen backbone, the method fixes hard deployment-time failures without full retraining. Real-robot results show strong recovery on difficult cases together with better preservation and lower training cost than retraining. These results support sparse human correction as a practical tool for robust robot manipulation. A longer version of this work with expanded technical details and experiments is available as a preprint *[ANONYMOUS]* and a video of the experiments can be found here: `https://tinyurl.com/4xwmvsxt`.

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
