# OpenReview forum: "Interactive Repair for Robust Generative Manipulation with Sparse Human Nudges"
_IEEE.org/ICRA/2026/Workshop/Manipulation_Robustness — ICRA 2026_

### Official Review · Reviewer_WEFt · 2026-05-13
**Review: Interactive Repair for Robust Generative Manipulation with Sparse Human Nudges**

**Rating:** 6
**Confidence:** 3

**Review:**

## Summary

The paper presents FlowCorrect, a lightweight repair mechanism for flow-matching manipulation policies. The idea is that when a deployed policy fails on edge cases, a human can provide sparse corrective nudges (relative pose adjustments via VR), and these get distilled into a small LoRA adapter + gating module on top of the frozen backbone. Evaluated on four tabletop tasks with a UR10, using only 8 base demos and 10 corrections per hard case.

## Strengths

1. The setup makes practical sense. A lot of real-world policy failures really are near-misses where the robot is almost in the right place, and it's wasteful to retrain the whole model for that. Framing repair as a sparse, relative correction rather than full re-demonstration is the right level of abstraction for this kind of problem.

2. Adding a gated correction to the flow field is a natural way to make the edit local, and the LoRA parameterization keeps the footprint small. Table II backs this up concretely: ~4x less GPU memory, roughly half the training time compared to retraining.

## Weaknesses

1. **Retraining actually wins on the paper's own headline metric.** Table I is framed as showing FlowCorrect's effectiveness on hard cases, but retraining matches or beats it on 3 out of 4 tasks — and by non-trivial margins on Pick-and-Place and Pouring (40/40 vs. 32/40 for both). The preservation argument hangs mostly on Insertion, where RT hurts overall ID success (Fig. 6), but that's one task. On the other three there's no clear evidence of regression from retraining. It would be more convincing to just say: retraining fixes hard cases better, but FlowCorrect is cheaper and less risky — rather than implying FlowCorrect is generally preferable.

2. **Testing on the same conditions you corrected is not that informative.** The ID-hard conditions in Table I were selected because the base policy fails there, corrections were collected on those exact conditions, and then success is reported on them again. That's basically train-set performance for the repair module. The real question — does the fix transfer to nearby unseen failures? — gets one OOD condition per task, and the answer is mixed (Pouring OOD is 2/10). More held-out failure conditions would strengthen the evaluation significantly.

3. **8 demos is a very weak base policy.** With only 8 demonstrations, the base policy is going to struggle on a lot of conditions that wouldn't be "hard" with a properly trained model. Every task shows 0/10 base success on all hard conditions — that's a policy that barely works outside its narrow training distribution. It's hard to tell whether FlowCorrect is solving a genuine deployment robustness problem or just patching an undertrained policy. Either show the method works on top of a stronger baseline (say 30–50 demos), or make an explicit argument for why 8 demos reflects a realistic deployment scenario.

## Questions

1. The gating module $g_\psi$ seems critical to the locality claim, but the paper doesn't explain how it's trained or what signal it uses to decide "this state is near a corrected region." Is it trained jointly with the LoRA weights? What inputs does it actually condition on?
2. If you scale the base policy to more demonstrations, do the hard cases persist? Or does FlowCorrect become unnecessary with a better-trained base?

---

### Decision · Program_Chairs · 2026-05-21

Accept